# Medical Radiation Safety and COVID-19 Knowledge and Awareness among UAE Residents: A Cross-Sectional Study

**DOI:** 10.3390/healthcare10071174

**Published:** 2022-06-23

**Authors:** Mustafa Alhasan, Mostafa Abdelrahman, Wijdan Alomaim, Mohammad Hasaneen, Qays Al-Horani, Suliman Salih, Mohd Nazmi Nordin

**Affiliations:** 1Radiologic Technology Program, Applied Medical Sciences College, Jordan University of Science and Technology, Irbid 22110, Jordan; maabdelrahman5@just.edu.jo; 2Radiography and Medical Imaging Department, Fatima College of Health Sciences, Abu Dhabi 3798, United Arab Emirates; wijdan.alomaim@fchs.ac.ae (W.A.); mohamed.hasaneen@fchs.ac.ae (M.H.); qays.alhorani@fchs.ac.ae (Q.A.-H.); suliman.salih@fchs.ac.ae (S.S.); mohd.nordin@fchs.ac.ae (M.N.N.); 3Diagnostic Imaging & Radiotherapy Program, Faculty of Health Sciences, Universiti Kebangsaan, Bangi 43600, Malaysia

**Keywords:** awareness, COVID-19, CT scan, chest X-ray, radiation safety

## Abstract

Radiologic examinations are valuable tools in the evaluation of COVID-19. A patient-centered care approach encourages patient involvement in decision-making related to their health management. Therefore, patients should have basic knowledge about their disease and its evaluation tools. Therefore, the purpose of this prospective study is to evaluate the public level of knowledge and awareness regarding COVID-19 and radiation safety in the UAE. **Methods**: A cross-sectional survey study was conducted using an online questionnaire (Google platform). The data collection instrument contained close-ended questions in both Arabic and English. The questions aimed to collect demographic information and to measure the level of knowledge and awareness of COVID-19 and radiation safety. The questionnaire was distributed online using different social media platforms. **Results**: A total of 1548 participants have completed the questionnaire; 84% were females and 16% were males. The participants’ average age was 24 years. Sixty-eight percent of the participants showed a high level of awareness of the COVID-19 pandemic, while most of the participants (51%) only showed a low level in the radiation safety awareness section. Factors such as Emirates of residence and passively receiving awareness information were shown to predict knowledge and awareness level. **Conclusions**: The UAE public was found to have a high level of knowledge and awareness about the COVID-19 disease. However, the same could not be said about radiation safety. More effort should be put towards raising the public’s knowledge and awareness about the risk of radiation in order to enable them to participate actively in decisions regarding the radiologic management of their disease.

## 1. Introduction

Coronavirus disease (COVID-19) first appeared in China in the city of Wuhan in December 2019 [1]. Since then, the disease was declared by the World Health Organization (WHO) as a Public Health Emergency of International Concern in January 2020, and then a pandemic on March 2020 [2]. COVID-19 is a pneumonia caused by severe acute respiratory syndrome coronavirus 2 (SARS-CoV-2), previously known as the 2019 novel coronavirus (2019-nCoV) [3]. The most common symptoms of the disease include fever, cough, and fatigue. More severe symptoms include high fever, severe cough, and shortness of breath [4]. As of the time of writing this manuscript, there have been over 520 million cases confirmed worldwide, with a mortality rate of 1.12% [5]. In the United Arab Emirates (UAE), there have been over 900,000 confirmed cases, with a mortality rate of only 0.25%.

SARS-CoV-2 is transmitted primarily through respiratory droplets, but aerosol and fomite transmission was also reported [6]. Therefore, social distancing, wearing masks, and hand hygiene are effective methods to protect against the spread of the disease [7].

A reverse transcriptase-polymerase chain reaction (RT-PCR) test performed for a nasopharyngeal swab is the primary method of diagnosing COVID-19 [8]. Chest computed tomography (CT) has also been proven to be valuable in the evaluation of the disease. In comparison with RT-PCR as a gold standard, chest CT has an overall sensitivity, specificity, positive predictive value, and negative predictive value of 87%, 46%, 69%, and 89%, respectively [9]. Typical CT findings in individuals with COVID-19 include ground-glass opacities and multiple areas of consolidation [8]. These radiological manifestations play a key role in tracking disease progression and assessing therapeutic effectiveness [10].

One downfall of CT scanning is radiation dose. In a study by Zhou et al. (2021) [11], chest CT was performed an average of four times on 550 COVID-19 patients during hospitalization, yielding a median effective does (ED) of 17.34 mSv (range, 2.05–53.39 mSv). This is more than 17 times the yearly limit on radiation exposure to a single member of the public [12].

Regarding the radiation safety, the International Commission on Radiological Protection (ICRP) have set three standards for radiation protection and safety: justification, optimization, and dose limitation. By reducing the exposure time, increasing the distance, and providing the patient with a protective shield, radiation safety can be achieved for both staff and patient [13].

Controlling the current pandemic requires an acceptable level of awareness and knowledge. The compliance of individuals can be improved if they are aware of the disease and the prevention methods [14,15], and it is safe to assume that this applies for COVID-19. There is an abundance of literature where the level of awareness among people about COVID-19 was evaluated. For example, a survey was distributed to investigate knowledge, attitudes, and practices about protecting the public against COVID-19 in China. About 96% were found not to visit crowded places, and 98% wore masks [16]. Additionally, another study was conducted to explore the awareness, attitudes, and practices of the general Pakistani population to COVID-19. The Pakistani general population was shown to have an overall positive attitude and act proactively against COVID-19 [17]. Moreover, a study attempted to assess the awareness, threat, symptoms, and prevention among people of India about COVID-19. This study has helped the government and people to understand and handle this coronavirus pandemic effectively [18].

Similarly, patients must be involved in the decision-making process in matters that concern their health management. This includes the decision to undertake radiologic examination. Therefore, it is desirable for the general public to have basic knowledge regarding the benefits and risks of imaging studies. Public awareness of radiation risks has been evaluated previously. For example, one study aimed to assess the level of patients’ awareness and knowledge regarding radiation and dosage along with the associated risks from computed tomography (CT) scan. A lack of awareness and knowledge about the use of ionizing radiation for diagnostic imaging was reported. Thus, imaging professionals are required to raise patients’ awareness [19].

Community healthcare education is an important factor for maintaining an acceptable quality of life. As we are still facing new COVID-19 waves, it is crucial for the public to demonstrate and maintain a good level of knowledge and awareness. In this study, public knowledge and awareness about COVID-19 and its radiologic management will be evaluated in the United Arab Emirates (UAE). This will help in identifying knowledge gaps in order to improve the overall awareness level.

## 2. Methods

This prospective cross-sectional online survey study was approved by the research ethical committee of the institution (INTSTF016RMI20) to be conducted among UAE residents. In this study, an online questionnaire was utilized and developed based on published information regarding COVID-19 awareness and on radiation safety, and was obtained from the WHO website and the relevant literature [20,21]. The questionnaire was created online using Google Forms and sent out to the participants to complete and submit electronically through social media platforms including Facebook and Twitter.

The questionnaire (English-Arabic version) was divided into 4 sections (Appendix A). The first section included information on the study and its purpose, and the consent form. The second section collected demographic information including gender, age, nationality, marital status, academic degree, job, work experience, work/study sector, and Emirate of residence. The third section included multiple choice questions (MCQs) regarding knowledge and awareness of the COVID-19 disease, including receiving information, pathogen, history, symptoms, risk factors, methods of prevention, and diagnosis. The fourth section collected information regarding knowledge and awareness of radiation safety on a Likert scale (1–5, “Not at all aware” to “Extremely aware”).

The questionnaire was piloted face-to-face with 10 people in order to confirm comprehension and clarity of the questions. The reliability coefficient measured using Cronbach’s alpha was 0.787.

A statistician was consulted to determine the sample size. It was decided to be 1200 in order to achieve a response rate of 50% with 95% confidence. Descriptive statistics were used to present demographics and knowledge scores including means, standard deviations (SD), and proportions. The mode was used to express the most frequent response. Knowledge score was first calculated as the percentage of correct answers, then divided into 3 levels: low (0–50%), moderate (51–75%), and high (76–100%). An independent sample Student’s t-test was used to compare the knowledge score between binary groups including gender (male vs. female), work sector (public vs. non-public), and Emirate of residence (inside capital vs. outside capital). A one-way analysis of variance test (ANOVA) was performed to test differences in the scores between groups of different degrees and jobs. Step-wise multiple linear regression analysis was used to find out predictors of the knowledge and awareness score (in percentage) for the radiation section. The explanatory variables included in the model were gender, age, work sector, Emirate of residence, academic degrees, jobs, and receiving information. Data were analyzed using IBM SPSS (IBM Corp., v.27.0. Armonk, NY, USA). A two-tailed *p*-value of 0.05 was considered to be significant.

## 3. Results

### 3.1. Participants’ Characteristics

A total of 1,654 participants agreed to participate. One hundred and six (6.4%) responses were excluded due to incomplete information. The final analysis included 1548 (94.6%) responses.

Most participants were females (n = 1299; 84%). The age of the participants ranged from 13 to 68, with a mean of 24 years old. Most of the participants held either a Bachelor’s degree (n = 796, 51.4%) or school education (n = 473, 30.6%), had no job (n = 1055; 68.2%), were working or studying in the public sector (n = 1132; 73.1%), and lived in the capital Abu Dhabi (n = 1074; 69.4%).

Most of the participants (n = 1517; 98%) received information regarding COVID-19 awareness, while only (n = 882, 57%) received information regarding radiation awareness.

Most participants (n = 1007; 65%) reported undergoing radiologic examinations at least once. The most visited department was the general X-ray (n = 768; 49.6%), while the least frequently visited department was fluoroscopy (n = 1; 0.1%), as shown in Table 1.

### 3.2. Knowledge and Awareness Scores

Regarding COVID-19, the average knowledge and awareness score was 85%. The majority of participants (n = 1056; 68%) scored in the high-level category, as shown in Figure 1. On the other hand, participants scored significantly lower in the radiation knowledge and awareness section, with a mean score of 50%. Indeed, the majority of participants (n = 796; 51.4%) scored in the low-level category, as shown in Figure 2.

Table 2 shows the radiation awareness level per question. The mode indicates the most selected awareness level (1 for the lowest, 5 for the highest). Ultrasound (US) and Magnetic resonance imaging (MRI) awareness received the lowest awareness level.

### 3.3. Participants’ Characteristics vs. Knowledge and Awareness Scores

Table 3 summarizes the knowledge and awareness scores for different groups.

#### 3.3.1. Gender

The female participants scored higher than their male counterparts in both COVID-19 and radiation awareness sections. There was a significant difference for the COVID-19 awareness section (*p* < 0.001). On the other hand, there was no significant difference for the radiation awareness section (*p* = 0.171).

#### 3.3.2. Academic Degree

According to the study level of the participants, the scores were similar among different degrees without significant difference (*p* > 0.998) for the COVID-19 awareness section. For the radiation awareness section, the doctorate level scored higher than the other degrees, while school level scored the lowest (*p* < 0.001).

#### 3.3.3. Job

According to the job title of the participants, medical jobs scored higher than the other categories in COVID-19 and radiation awareness sections, and non-medical jobs scored the lowest. There was as significant difference for both sections’ scores (*p* < 0.001).

#### 3.3.4. Work/Study Sector

The public sector participants scored higher than non-public participants. There was a significant difference for COVID-19 awareness (*p =* 0.018), while there was no significant difference for the radiation scores awareness section (*p =* 0.622).

#### 3.3.5. Emirate of Residence

Residents outside the capital scored similar results to the capital’s residents for the COVID-19 awareness section without significant difference **(***p* = 0.348). On the other hand, the capital’s residents scored higher than residents outside the capital, and the difference was significant **(***p* < 0.001) for the radiation scores awareness section.

### 3.4. Predictor Factors for the Radiation Awareness Section

The linear regression test showed that the radiation knowledge and awareness score could be partially predicted from the address (*p* = 0.032) and the receiving educational information (*p* < 0.001). The score appeared independent of all other factors.

## 4. Discussion

The first case of COVID-19 appeared in the UAE in January 2020. During the first wave of the outbreak, the number of cases steadily increased, creating fear and panic among the general public. Often, suspected COVID-19 and the positive cases will undergo radiologic examination, most commonly chest X-rays or CT scans. Therefore, the purpose of this prospective study was to evaluate the level of COVID-19 awareness towards the emerging COVID-19 disease and radiation safety among the UAE community.

Sixty eight percent of the participants showed a high-level of knowledge and awareness regarding COVID-19. This could be due to the fact that 98% of the participants had received awareness information about the disease. This is in agreement with an earlier report from Saudi Arabia, where participants achieved a knowledge score of 95% [22].

Some participants’ characteristics were significantly associated with the level of awareness. The female participants scored higher than the males in both COVID-19 (81% females vs. 77% males) and radiation awareness (54% females vs. 52% males) sections. This is in an agreement with a study which indicated that women are more likely to perceive the pandemic as a very serious health problem, and to agree and comply with restraining measures comparing to men [23].

People with higher levels of education were more knowledgeable (Doctorate, 69%) compared to other categories regarding radiation awareness (Master’s, 59%; Bachelor’s, 58%; Diploma, 52%; and School, 51%). As anticipated, participants with a medical job category scored higher than the other jobs for both COVID-19 and radiation awareness sections (85% and 78%, respectively). Similarly, a study was conducted among health profession students with different academic levels including undergraduate, interns, and postgraduate students. The results indicated statistically significant differences in knowledge with postgraduate students having the highest mean scores, followed by interns and undergraduate students [24]. Additionally, another study highlighted the importance of level of education in enhancing knowledge and attitudes, which leads to successful hygienic practice performance during COVID-19 [25]. Moreover, a Syrian study reported that age, education, level of education and occupation were the only significant factors that improved the level of COVID-19 awareness [26].

In contrast to radiation awareness, education level showed no significant difference between different study levels regarding COVID-19 awareness, which could be due to the fact that COVID-19 is a pandemic, involving local, regional and international awareness campaigns and news, which targeted and educated all society members of different ages.

Regarding the radiation awareness section, ultrasound and MRI awareness questions showed the lowest awareness level, while X-ray awareness was higher. This could be due to the high demand on X-ray departments, as most of the participants (50%) reported that the X-ray department was the most visited department.

Overall, UAE residents showed a high-level of awareness of COVID-19 (85%). This is concurrent with a study that was conducted in USA. The results showed that 83% of people knew that COVID-19 can be transmitted from contaminated surfaces, and 87% knew the three common symptoms (fever, cough, difficulty in breathing) of COVID-19 [27]. In contrast, another study of adults with chronic conditions indicated a lack of critical knowledge about COVID-19 [28].

On the other hand, our study showed that 51% of the respondents scored in the low awareness level regarding radiation awareness. This is similar to an Italian study where the majority of the patients (56%) did not know which modality uses ionizing radiation [29].

The current study has some limitations. Regarding the gender distribution, female participants were more active and willing to participate than males, though the survey was shared with both genders across different ages through different social media platforms. This led to a higher female proportion than male participants. This might be explained by the fact that females use social media more often than males [30]. Additionally, the study attracted more participation from the young population (the mean age was 24 years old) than the elder. The young age of the participants is a reflection of their higher activity on social media compared to the elders.

In addition, the number of doctorate level participants was low in comparison to the other degrees. Nevertheless, this study has provided an insight into the community awareness level and identified the lack of knowledge’s domains, and it can help other investigators with their future studies within UAE or other countries.

## 5. Conclusions

In this study, the overall knowledge and awareness level of the participants regarding COVID-19 was high. This was demonstrated by the average score of 85% and by the fact that the majority of the participants (68%) scored in the high-level category (76–100%). This is a reflection of the UAE government’s efforts towards reducing the spread of the pandemic through awareness campaigns that targeted the whole population. Having a high level of knowledge and awareness about COVID-19 makes an important contribution toward enabling patients to participate actively in the decision-making process regarding the radiologic management of their disease. However, this favourable aspect is counteracted by the low level of knowledge and awareness about radiation safety. Most of the participants (51.4%) scored in the low-level category, with and overall average score of only 50%. Poor knowledge and awareness in radiation safety was especially true for recognizing which imaging modalities use harmless forms of radiation, including US and MRI. In order to support patients’ right to autonomy and self-determination during radiologic management, efforts should be made towards raising their knowledge and awareness about the risks and benefits of different radiologic options.

## Figures and Tables

**Figure 1 healthcare-10-01174-f001:**
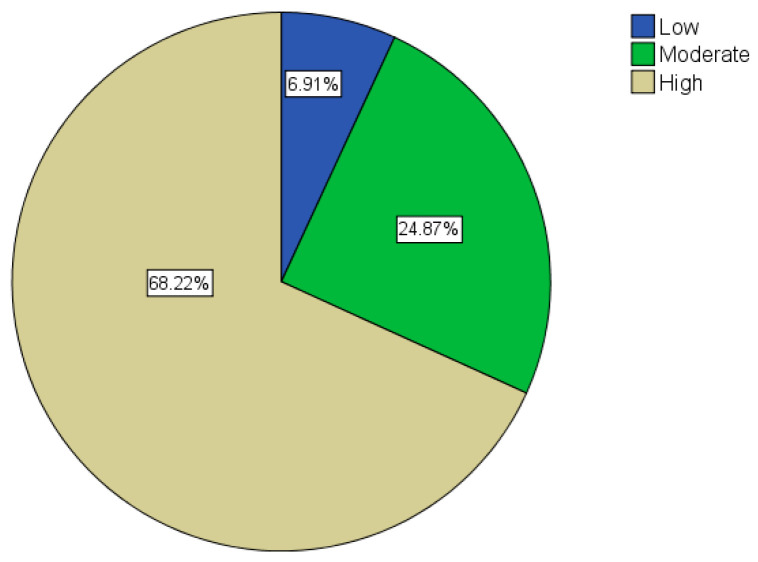
COVID-19 awareness categories.

**Figure 2 healthcare-10-01174-f002:**
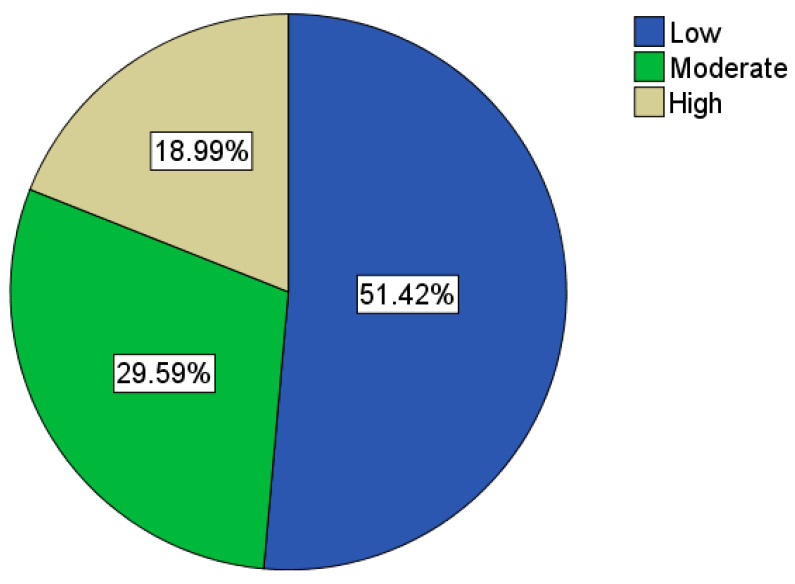
Radiation awareness categories.

**Table 1 healthcare-10-01174-t001:** Frequency of visited imaging departments.

Department	Frequency	Percent
Computed tomography (CT)	65	4.2
Dental	74	4.8
DEXA	4	0.3
Fluoroscopy	1	0.1
General X-ray	768	49.6
Mammography	16	1.0
Magnetic resonance imaging (MRI)	44	2.8
Portable/Mobile	4	0.3
Ultrasound (US)	31	2.0
None	541	34.9
Total	1548	100.0

**Table 2 healthcare-10-01174-t002:** Likert scale results for radiation awareness questions.

	Ionizing Radiation Exposure Can Induce Cancer	X-ray is a Harmful Form of Radiation	Ultrasound Imaging Uses a Safe Form of Radiation	MRI Uses a Safe Form of Radiation
**Subjects**	1548	1548	1548	1548
**Mode**	3	3	1	1

**Table 3 healthcare-10-01174-t003:** Scores of COVID-19 and radiation awareness sections of different groups.

		**Gender**	**Subjects**	**Mean**	**Std. Deviation**	***p* value**
	COVID-19 awareness score	Female	1299	0.8136	0.14960	< 0.001
		Male	249	0.7758	0.15921	
	Radiation awareness score	Female	1299	0.5431	0.21867	0.171
		Male	249	0.5221	0.24041	
		**Degree**	**Subjects**	**Mean**	**Std. Deviation**	***p* value**
	COVID-19 awareness scores	School	473	0.8087	0.15026	0.998
		Diploma	210	0.8048	0.15631	
		Bachelor	796	0.8074	0.15101	
		Master	64	0.8099	0.16229	
		Doctorate	5	0.8000	0.13944	
	Radiation awareness scores	School	473	2.5513	1.03110	< 0.001
		Diploma	210	2.6071	1.09334	
		Bachelor	796	2.7864	1.14235	
		Master	64	2.9414	1.21478	
		Doctorate	5	3.4500	1.16458	
		**Job**	**Subjects**	**Mean**	**Std. Deviation**	***p* value**
	COVID-19 Awareness score	No job	1055	0.8145	0.15086	< 0.001
		Housewife	75	0.8156	0.12428	
		Non-Medical	320	0.7688	0.15944	
		Medical	98	0.8520	0.13082	
	Radiation awareness	No job	1055	0.5424	0.21809	< 0.001
		Housewife	75	0.5153	0.19154	
		Non-Medical	320	0.4634	0.19300	
		Medical	98	0.7796	0.20831	
		**Work/study sector**	**Subjects**	**Mean**	**Std. Deviation**	***p* value**
COVID-19 Awareness score		Non-Public	208	0.7812	0.16894	0.018
		Public	1132	0.8086	0.15041	
Radiation Awareness score		Non-Public	208	0.5284	0.21872	0.622
		Public	1132	0.5367	0.22398	
		**Address**	**Subjects**	**Mean**	**Std. Deviation**	***p* value**
COVID-19 Awareness score		Outside Capital	474	0.8129	0.13642	0.348
		Abu Dhabi Capital	1074	0.8051	0.15807	
Radiation Awareness score		Outside Capital	474	0.4945	0.19831	< 0.001
		Abu Dhabi Capital	1074	0.5597	0.22945	

## Data Availability

The datasets used and analyzed during the current study are available from the corresponding author upon reasonable request.

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
