# Peer review of "Medical Radiation Safety and COVID-19 Knowledge and Awareness among UAE Residents: A Cross-Sectional Study"

_healthcare, 2022, doi:10.3390/healthcare10071174_

Round 1
Reviewer 1 Report
The mansucript requires major edition. In the present form can not be suitable for being published in Healthcare or any other scientific journal. Originality should be highlighted, and precise details of methodology must be detailed. Questions should be clear and unbiased, the statitical tests and programs to analyze must be checked and clearly explained.
Discussion should be deeper regarding previous exploration in this field, albeit if it was not in pandemic time. References number should be increased.
The conclusions are not supported clearly by the results or discussion. It is considered enough for the authors that UAE residents consider dangerous the application of X-ray?
Author Response
- The manuscript requires major edition. In the present form cannot be suitable for being published in Healthcare or any other scientific journal. Originality should be highlighted, and precise details of methodology must be detailed. Questions should be clear and unbiased, the statistical tests and programs to analyze must be checked and clearly explained.
- The authors would like to thank the reviewer for the valuable comments and suggestions. The entire manuscript has been revised and most of the sections were amended to improve the scientific and the English writing style of the manuscript. The last section of the introduction was modified to clarify the originality of the paper. Additionally, the methodology section was expanded to include more details. Regarding the research questions, the questionnaire was piloted face-to-face with 10 people in order to confirm comprehension and clarity of the questions. The reliability coefficient measured using Cronbach's alpha was 0.787. The statistical tests were explained as required. Data were analyzed using IBM SPSS (IBM Corp., v.27.0. Armonk, NY).
- Discussion should be deeper regarding previous exploration in this field, albeit if it was not in pandemic time. References number should be increased.
- More information and references have been added to elaborate more on the research topic. Ten more references were added to increase the total number of references to 30.
- The conclusions are not supported clearly by the results or discussion. It is considered enough for the authors that UAE residents consider dangerous the application of X-ray?
- The conclusion section was modified. More information was added to summarize the paper outcomes in the light of evidence from the results and discussion.
Reviewer 2 Report
Article discussed the awareness of COVID-19 among UAE residents and questionnaire was conducted to test it. This study is novel. Following suggestions need to be incorporated.
-Line 34, on February, year to be provided.
-Discussion must include reason for lowest scores of housewives (5%) etc. Similarly there must be reason and discussion for other participants results.
-Line 176, 68% at start of line must be Sixty eight percent.
-Conclusions should be extended including some of the numbers and results quoted in the results.
Author Response
- Article discussed the awareness of COVID-19 among UAE residents and questionnaire was conducted to test it. This study is novel. Following suggestions need to be incorporated.
- The authors would like to thank the reviewer for the valuable comments and suggestions. The entire manuscript has been revised and most of the sections were amended to improve the scientific and the English writing style of the manuscript.
- -Line 34, on February, year to be provided.
- The requested information was provided.
- -Discussion must include reason for lowest scores of housewives (5%) etc. Similarly there must be reason and discussion for other participants results.
- The authors would like to clarify the 5% is the percentage of housewives who participated in the study, not a score. More information and references have been added to elaborate more on the research topic. Ten more references were added to increase the total number of references to 30.
- -Line 176, 68% at start of line must be Sixty eight percent.
- The comments were addressed as suggested.
- -Conclusions should be extended including some of the numbers and results quoted in the results.
- The conclusion section was modified to discuss the research findings as suggested.
Reviewer 3 Report
It is a cross sectional questionnaire-based study on medical radiation safety and COVID-19 knowledge and awareness among UAE residents. The methodology chosen was correct and conclusions are driven by the results.
Some minor comments: Was the group representative? On which grounds? as you stated in the limitations, these were mainly women (84%) and young people (mean 24 yrs). Please comment low age in the limitations.
Please explain ‘UAE’ country name in the introduction and then only use it accordingly.
Questionnaire should be available as supplementary material.
Author Response
- It is a cross sectional questionnaire-based study on medical radiation safety and COVID-19 knowledge and awareness among UAE residents. The methodology chosen was correct and conclusions are driven by the results.
- The authors would like to thank the reviewer for the valuable comments and suggestions. The entire manuscript has been revised and most of the sections were amended to improve the scientific and the English writing style of the manuscript.
- Some minor comments: Was the group representative? On which grounds? as you stated in the limitations, these were mainly women (84%) and young people (mean 24 yrs). Please comment low age in the limitations.
- The comments were addressed in the limitation section of the discussion. This is a reflection for the higher prevalence of social media use among females and the younger population. Sample size justification was added to the material and methods section.
- Please explain ‘UAE’ country name in the introduction and then only use it accordingly.
- The comments were addressed as suggested.
- Questionnaire should be available as supplementary material.
- The questionnaire was added as supplementary material as suggested.
Round 2
Reviewer 1 Report
Please check the entire manuscript to avoid grammar and typo mistakes. All language should be in scientific expressions.
'900k'
'predicor'
'asscoiated'
Conclusions must be clearly supported by (and linked to) results and discussion.
Author Response
- Please check the entire manuscript to avoid grammar and typo mistakes. All language should be in scientific expressions.
'900k'
'predicor'
'asscoiated'
- The authors would like to thank the reviewer for the valuable comments. The entire manuscript was revised and corrected for grammar, typo mistakes, and proper scientific language.
- Conclusions must be clearly supported by (and linked to) results and discussion.
- The authors would like to thank the reviewer for the valuable comments. Conclusion was updated as per the reviewer recommendation.